# Robot makes hamburger by variable speed motion-copying system

Koki Yamane[1], Sho Sakaino[1] and Toshiaki Tsuji[2]

*Abstract*— Motion-copying systems are useful for teaching robots complex motions, and this method can realize faster or slower motions than human demonstrations. However, it has not been demonstrated for complex motions involving sliding the fingers under an object, flipping a grasped object, or grasping a multiplicity of flexible objects at the same time. In this study, we conduct evaluation tests for an assembling hamburger task, which includes the above-mentioned challenging motions, to verify the availability of a variable-speed motion-copying system for the manipulator's complex tasks. As a result, we could show the availability of the motion-copying system used at a speed of 1x to 3x for the assembling hamburger task.

## I. Introduction

Motion-copying systems [1] , based on 4-channel bilateral control, are a useful way to teach robots complex motions, including force information. Moreover, this method can realize faster or slower motion than human demonstrations [1]. However, it has not been demonstrated for complex motions involving sliding fingers under an object, flipping a grasped object, or grasping a multiplicity of flexible objects at the same time. In this study, we conduct evaluation tests for an assembling hamburger task, which includes the above-mentioned challenging motions, to verify the availability of a variable-speed motion-copying system for the manipulator's complex tasks. As a result, we could show the availability of the motion-copying system used at a speed of 1x to 3x for the assembling hamburger task.

## II. Robot System

### A. Manipulator

CRANE-X7, a manipulator manufactured by RT corporation, was employed. The manipulator exhibits seven degrees of freedom, while the gripper exhibits one degree of freedom, thereby providing a total of eight degrees of freedom. We replaced the robot's gripper with a cross-structure hand [2] without finger holes. The overview of the system is presented in Fig. 1.

## III. Control System

### A. Position and Force Hybrid Controller

Each axis was controlled using a position and force hybrid controller. The block diagram of the controller is presented in Fig. 2.

The proportional-derivative (PD) control was applied to the position, and the proportional (P) control was applied

[1]The authors are with the Graduate School of Systems and Information Engineering, University of Tsukuba, 1-1-1 Tennodai, Tsukuba, Ibaraki 305-8577, Japan yamane.koki.td@alumni.tsukuba.ac.jp

[2]The author is with the Graduate School of Science and Engineering, Saitama University, Saitama 338-8570, Japan

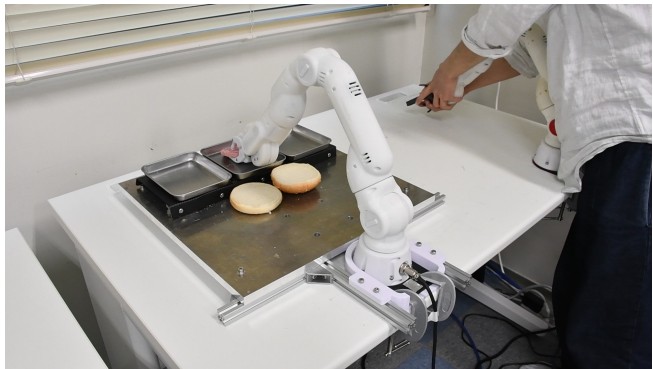

(a) 4-channel bilateral control

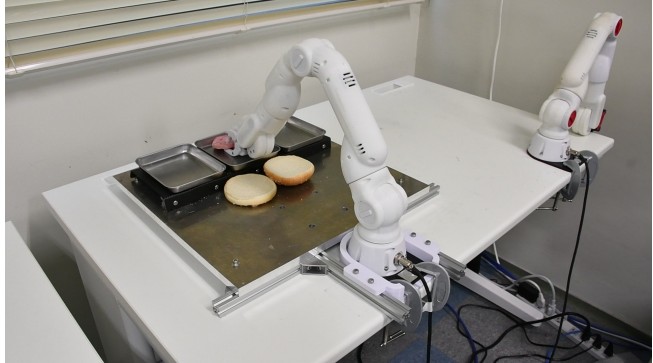

(b) Motion-copying system

Fig. 1: Overview of experiment

to the force. The disturbance torque of each joint $\hat{\tau}_{dis}$ was calculated and suppressed using a disturbance observer (DOB) [3], and the torque response value $\hat{\tau}_{res}$ was estimated using a reaction force observer (RFOB) [4]. The control frequency was set to 500 Hz.

Here, the dynamics of the manipulator include viscous friction and reaction force can be expressed as below:

$$\boldsymbol{\tau}^{ref} = \boldsymbol{M}(\boldsymbol{\theta})\ddot{\boldsymbol{\theta}} + (\boldsymbol{h}(\boldsymbol{\theta}, \dot{\boldsymbol{\theta}}) + \boldsymbol{D})\dot{\boldsymbol{\theta}} + \boldsymbol{G}(\boldsymbol{\theta}) + \boldsymbol{\tau}^{res} \quad (1)$$

$$= (\boldsymbol{M}_n + \Delta\boldsymbol{M}(\boldsymbol{\theta}))\ddot{\boldsymbol{\theta}} + (\boldsymbol{h}(\boldsymbol{\theta}, \dot{\boldsymbol{\theta}}) + \boldsymbol{D})\dot{\boldsymbol{\theta}} + \boldsymbol{G}(\boldsymbol{\theta}) + \boldsymbol{\tau}^{res} \quad (2)$$

$$= \boldsymbol{M}_n\ddot{\boldsymbol{\theta}} + \boldsymbol{\tau}^{dis} \quad (3)$$

$$\Delta\boldsymbol{M} = \boldsymbol{M}(\boldsymbol{\theta}) - \boldsymbol{M}_n \quad (4)$$

$$\boldsymbol{\tau}^{dis} = \Delta\boldsymbol{M}(\boldsymbol{\theta})\ddot{\boldsymbol{\theta}} + (\boldsymbol{h}(\boldsymbol{\theta}, \dot{\boldsymbol{\theta}}) + \boldsymbol{D})\dot{\boldsymbol{\theta}} + \boldsymbol{G}(\boldsymbol{\theta}) + \boldsymbol{\tau}^{res} \quad (5)$$

where $\boldsymbol{\theta}$, $\dot{\boldsymbol{\theta}}$, $\ddot{\boldsymbol{\theta}}$, and $\boldsymbol{\tau}$ denote the angle, angular velocity, angular acceleration, and torque of each joint, respectively, and the superscripts "res," "ref," and "dis" denote the response, reference, and disturbance values, respectively. In addition,

TABLE I: Parameters of the position and force hybrid controller

| task | $g_v$ [rad/s] | $g_{dob}$ [rad/s] | $M_n$ [Nm·s2/rad] | $D$ [Nm·s/rad] | $G(\theta)$ [Nm] | $K_p$ | $K_d$ | $K_f$ |
|---|---|---|---|---|---|---|---|---|
| joint 0 | 15 | 15 | 0.0123 | 0.0501 | 0.0 | 256 | 40 | 0.7 |
| joint 1 | 15 | 15 | 0.1130 | 0.0000 | $2.0945 \cdot \sin(\theta_1^{res}) + 1.1505 \cdot \sin(\theta_1^{res} + \theta_3^{res})$ | 196 | 28 | 0.7 |
| joint 2 | 20 | 20 | 0.0120 | 0.2420 | 0.0 | 961 | 66 | 0.0 |
| joint 3 | 20 | 20 | 0.0400 | 0.0000 | $1.183 \cdot \sin(\theta_1^{res} + \theta_3^{res})$ | 144 | 24 | 1.0 |
| joint 4 | 20 | 20 | 0.0057 | 0.0400 | 0.0 | 289 | 34 | 0.8 |
| joint 5 | 20 | 20 | 0.0066 | 0.0391 | 0.0 | 324 | 36 | 1.0 |
| joint 6 | 20 | 20 | 0.0063 | 0.0500 | 0.0 | 144 | 24 | 0.8 |
| gripper | 20 | 20 | 0.0069 | 0.0210 | 0.0 | 324 | 36 | 1.0 |

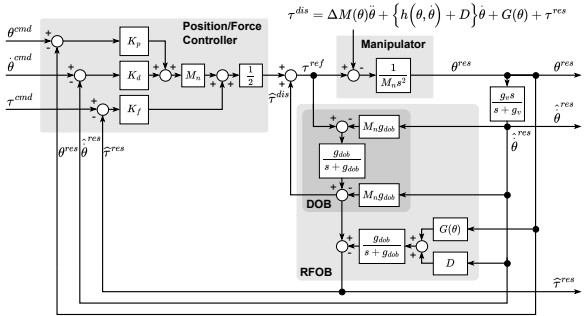

Fig. 2: Block diagram of position and force hybrid controller

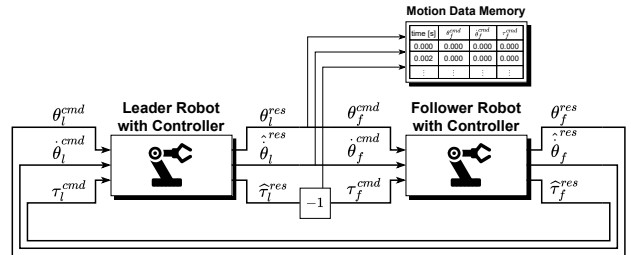

Fig. 3: Block diagram of 4-channel bilateral control

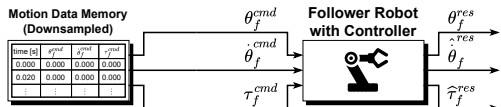

Fig. 4: Block diagram of motion-copying system

$M(\theta)$, and $M_n$ are uncertain and nominal inertias, $D$ is the coefficient of viscous friction, $h(\theta, \dot{\theta})\dot{\theta}$ is centrifugal and Coriolis forces, $G(\theta)$ is gravity, respectively.

The torque reference value was calculated as

$$\hat{\dot{\theta}}^{res} = \frac{g_v s}{s + g_v}\theta^{res}, \quad \dot{\theta}^{cmd} = \frac{g_v s}{s + g_v}\theta^{cmd} \tag{6}$$

$$\hat{\tau}^{dis} = \frac{g_{dob}}{s + g_{dob}}(\tau^{ref} - M_n s \hat{\dot{\theta}}^{res})$$
$$= \frac{g_{dob}}{s + g_{dob}}(\tau^{ref} + M_n g_{dob}\hat{\dot{\theta}}^{res}) - M_n g_{dob}\hat{\dot{\theta}}^{res} \tag{7}$$

$$\hat{\tau}^{res} = \hat{\tau}^{dis} - \frac{g_{dob}}{s + g_{dob}}(D\hat{\dot{\theta}}^{res} + G(\theta)) \tag{8}$$

$$\tau^{ref} = \frac{M_n}{2}(K_p + K_d \frac{g_v s}{s + g_v})(\theta^{cmd} - \theta^{res})$$
$$+ \frac{1}{2}K_f(\tau^{cmd} - \hat{\tau}^{res}) + \hat{\tau}^{dis}$$
$$= \frac{M_n}{2}K_p(\theta^{cmd} - \theta^{res}) + \frac{M_n}{2}K_d(\dot{\theta}^{cmd} - \hat{\dot{\theta}}^{res})$$
$$+ \frac{1}{2}K_f(\tau^{cmd} - \hat{\tau}^{res}) + \hat{\tau}^{dis} \tag{9}$$

where $g_v$, and $g_{dob}$ are cut-off frequencies of the low-pass filters, and $K_p$, $K_d$, and $K_f$ represent the control gains of position, velocity, and force controls, respectively. These parameters are set to the values shown in Table I. Here, the superscripts "cmd" denote the command values, and ˆdenotes the estimated value.

### B. 4-Channel Bilateral Control

To collect demonstration data, we employed 4-channel bilateral control [5], a system comprising two robots: a leader directly manipulated by a human and a follower of the leader.

The block diagram of the 4-channel bilateral control is shown in Fig. 3.

The system controls the position and torque of the two robots to synchronize them. The target of 4-channel bilateral control are defined as below:

$$\theta_l^{res} - \theta_f^{res} = 0 \tag{10}$$
$$\tau_l^{res} + \tau_f^{res} = 0 \tag{11}$$

where subscripts $l$ and $f$ represent the leader and follower, respectively. Therefore, the angle, angular velocity, and torque command values for the position and force hybrid controller are defined as

$$\theta_f^{cmd} = \theta_l^{res}, \theta_l^{cmd} = \theta_f^{res} \tag{12}$$
$$\dot{\theta}_f^{cmd} = \dot{\theta}_l^{res}, \dot{\theta}_l^{cmd} = \dot{\theta}_f^{res} \tag{13}$$
$$\tau_f^{cmd} = -\tau_l^{res}, \tau_l^{cmd} = -\tau_f^{res} \tag{14}$$

respectively.

To use in the motion-copying system, $\theta_f^{cmd}$, $\dot{\theta}_f^{cmd}$, and $\tau_f^{cmd}$ of each time step are saved to the motion data memory.

### C. Motion-copying System

In motion-copying systems, human demonstrations are collected by using 4-channel bilateral control, while the collected motions are reproduced by using motion-reproduction systems as shown in Fig. 4. When we use the motion-copying system, we use motion data memory instead of the leader robot of the 4-channel bilateral control system. $\theta_f^{cmd}$, $\dot{\theta}_f^{cmd}$,

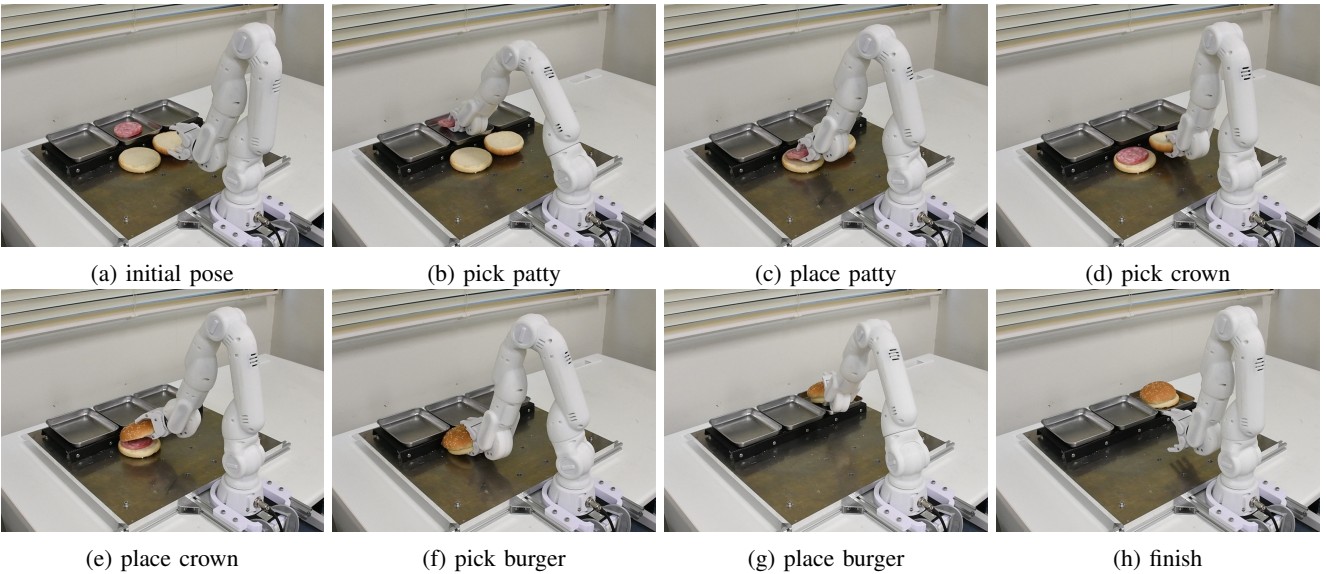

| (a) initial pose | (b) pick patty | (c) place patty | (d) pick crown |

| (e) place crown | (f) pick burger | (g) place burger | (h) finish |

Fig. 5: Task snapshot

and $\boldsymbol{\tau}_f^{cmd}$ of each time step are loaded from motion data memory, and input to the position and force hybrid controller of the follower robot. The control frequency of the position and force hybrid controller was set to 500 Hz, and the frequency of the command value update was set to 50 Hz. We change motion speed by downsampling saved data on motion data memory.

## IV. Experiments

### A. Task Design

In this study, the task was to assemble a hamburger. The entire task flow is shown in Fig. 5. The experimental environment consists of a step and three trays, with the patties placed on the trays and the buns on the bottom of the step. The task is divided into six steps. First, lift the patty placed on the center tray (pick patty, Fig. 5b). Place it on the bottom bun (place patty, Fig. 5c), then lift the upper bun (pick crown, Fig. 5d), and place it on top (place crown, Fig. 5e). Lift the finished burger (pick burger, Fig. 5f), and place it on the tray on the right side of the image (place burger, Fig. 5g).

### B. Results

We conduct evaluation tests for 1x to 4x speed motion-copying systems 5 times each. The success rates are presented in Table II. The cases using 1x and 2x speed were all successful, and 3x speed cases were 80% successful. On the other hand, all the cases using 4x speed failed. In the 4x speed case, the trajectory of the robot's movement was very smooth, and the robot couldn't move precisely to the pick pose and place pose because they were at the edge of the trajectory. The failed case of the 3x speed trial looks to have the same issue. This is thought to be due to the hardware limitation and the smoothing by downsampling.

TABLE II: Success rate

| speed | 1x | 2x | 3x | 4x |
|---|---|---|---|---|
| success rate [%] | **100** (5/5) | **100** (5/5) | 80 (4/5) | 0 (0/5) |

## V. Conclusions

We conduct evaluation tests for the assembling hamburger task to verify the availability of a variable-speed motion-copying system for the manipulator's complex tasks. As a result, we could show the availability of the motion-copying system used from 1x to 3x speed. In contrast, the 4x speed case couldn't succeed due to not only the hardware limitation but also the smoothing by downsampling. A future issue is to perform time scaling by signal processing that takes into account the attenuation of high-frequency components due to downsampling.

## Acknowledgment

This study is supported by the Japan Society for the Promotion of Science through a Grant-in-Aid for Scientific Research (B) under Grant 21H01347.

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
