# OpenReview forum: "Robot makes hamburger by variable speed motion-copying system"
_IEEE.org/2024/ICRA/Workshop/CookingRobot — CookingRobot2024 Poster_

### Official Review · Reviewer_HyvL · 2024-04-08
**The review of "Robot makes hamburger by variable speed motion-copying system"**

**Rating:** 9
**Confidence:** 5

**Review:**

*Major Contribution of the Paper:

The paper evaluated the motion-copying system which enables teaching robots complex motions with variable speed.
They demonstrate the complex task of making hamburgers, which involves challenging operations like sliding fingers under buns and patties, flipping the buns, and grasping multiple flexible items simultaneously.


*Major comments:

The research showcases promising advancements in motion-copying systems for real-world applications. The paper demonstrated a complicated task with different speeds, which provides valuable experimental results. Expanding the discussion on broader implications and potential applications beyond the specific hamburger assembly task could enhance the paper's overall significance.

*Video:

-	Impressive. The movement of the robot is very smooth and fast.


*Technical Accuracy:

-	The methodology and experimental setup are well-described, which provides a clear understanding of the technology.

-	While the paper mentions challenges faced during higher speed replication (4x), it lacks an in-depth analysis of these challenges and potential solutions. Further discussion could enhance the paper's quality.

---

### Official Review · Reviewer_E3v9 · 2024-04-11
**The review of "Robot makes hamburger by variable speed motion-copying system"**

**Rating:** 7
**Confidence:** 4

**Review:**

This paper proposes a variable speed motion-copying systems for complex motions such as making hamburger.
The results show that motion-copying system can used at a speed of 1x to 3x for the assembling hamburger task, but 4x was failed.

Major Comment
* There is a certain value in successfully completing complex tasks such as the hamburger task with a motion-copying system that can vary its speed.

Video
* It's a very good video, but it would be even better to include some failed attempts, like 3x or 4x.